# Pvr expression regulators in equilibrium signal control and maintenance of *Drosophila* blood progenitors

Bama Charan Mondal[1], Jiwon Shim[1,2], Cory J Evans[1]*, Utpal Banerjee[1,3,4,5]*

[1]Department of Molecular, Cell and Developmental Biology, University of California, Los Angeles, Los Angeles, United States; [2]Department of Life Science, Hanyang University, Seoul, Republic of Korea; [3]Department of Biological Chemistry, University of California, Los Angeles, Los Angeles, United States; [4]Molecular Biology Institute, University of California, Los Angeles, Los Angeles, United States; [5]Eli and Edythe Broad Center of Regenerative Medicine and Stem Cell Research, University of California, Los Angeles, Los Angeles, United States

**Abstract** Blood progenitors within the lymph gland, a larval organ that supports hematopoiesis in *Drosophila melanogaster*, are maintained by integrating signals emanating from niche-like cells and those from differentiating blood cells. We term the signal from differentiating cells the 'equilibrium signal' in order to distinguish it from the 'niche signal'. Earlier we showed that equilibrium signaling utilizes Pvr (the *Drosophila* PDGF/VEGF receptor), STAT92E, and adenosine deaminase-related growth factor A (ADGF-A) (*Mondal et al., 2011*). Little is known about how this signal initiates during hematopoietic development. To identify new genes involved in lymph gland blood progenitor maintenance, particularly those involved in equilibrium signaling, we performed a genetic screen that identified *bip1* (*bric à brac interacting protein 1*) and *Nucleoporin 98* (*Nup98*) as additional regulators of the equilibrium signal. We show that the products of these genes along with the Bip1-interacting protein RpS8 (Ribosomal protein S8) are required for the proper expression of Pvr.

**\*For correspondence:** cjevans@ucla.edu (CJE); banerjee@mbi.ucla.edu (UB)

**Reviewing editor**: Benjamin Ohlstein, Columbia University Medical Center, United States

## Introduction

Similar to vertebrates, blood cell differentiation in *Drosophila* is regulated in multiple hematopoietic environments, which include the head mesoderm of the embryo (*Tepass et al., 1994*; *Lebestky et al., 2000*; *Milchanowski et al., 2004*), the specialized, tissue-associated microenvironments of the larval periphery (e.g, body wall hematopoietic pockets) (*Markus et al., 2009*; *Makhijani et al., 2011*), and the larval lymph gland, an organ dedicated to the development of blood cells that normally contribute to the pupal and adult stages (*Rizki, 1978*; *Shrestha and Gateff, 1982*; *Lanot et al., 2001*; *Jung et al., 2005*). Understanding how blood cell development is regulated in the lymph gland is the primary goal underlying the work presented here. Differentiating blood cells (hemocytes) of the lymph gland are derived from multipotent progenitors (*Jung et al., 2005*; *Mandal et al., 2007*; *Martinez-Agosto et al., 2007*). These blood progenitors readily proliferate during the early growth phases of lymph gland development, which is followed by a period in which many of these cells slow their rate of division and are maintained without differentiation in a region termed the medullary zone (MZ, *Figure 1*) (*Jung et al., 2005*; *Mandal et al., 2007*). During the same period, other progenitor cells begin to differentiate along the peripheral edge of the lymph gland to give rise to a separate cortical zone (CZ) (*Jung et al., 2005*). How progenitor cell maintenance and differentiation are regulated during the course of lymph gland development has become a major area of exploration in recent years, and several different signaling pathways have been identified that maintain progenitor cells through

**eLife digest** Progenitor cells are cells that can either multiply to make new copies of themselves or mature into different specialized cell types—such as blood cells. In the fruit fly *Drosophila*, new blood cells are formed in several different locations, including in an organ called the lymph gland.

In 2011, researchers found that the fate of blood progenitor cells within the lymph gland is controlled by signals from two nearby sources—one from specialized, supportive ('niche') cells and the other from maturing blood cells. The signal from the maturing blood cells ensures that the relative amounts of progenitor and maturing blood cells are kept in the right balance. As a result, this signaling process has been called 'equilibrium signaling'.

Questions remain as to how equilibrium signaling is regulated, and how it interacts with signals from the niche. To investigate this, Mondal et al.—including some of the researchers involved in the 2011 work—used various genetic techniques to create *Drosophila* larvae in which the tissues that become blood cells are made visible with fluorescent proteins. This meant that these tissues could be examined in live, whole animals by using a microscope. Mondal et al. then searched for the *Drosophila* genes involved in generating new blood cells in the lymph gland—particularly those involved in equilibrium signaling. This was done by switching on and off hundreds of genes, one by one, in the lymph gland, and any genes that caused changes to the generation of new blood cells were then investigated further.

Following these investigations, Mondal et al. focused on three genes—and when each of these genes was switched off in maturing blood cells, the result was that fewer progenitor cells remained in the lymph gland. This effect was not seen when the genes were switched off in the progenitor or the niche cells, which suggested that the genes are likely to be components of the equilibrium signaling pathway. Switching off these genes in maturing blood cells also dramatically reduced the levels of a protein called Pvr, a key equilibrium signaling protein known from the 2011 study and an important player in blood cell development in several species.

How the newly identified genes actually control Pvr protein levels to maintain proper equilibrium signaling in the lymph gland remains to be explored. However, this work provides a basis for investigating the role of related genes in blood cell development in vertebrate systems, namely humans.

the larval stages (*Lebestky et al., 2003*; *Mandal et al., 2007*; *Owusu-Ansah and Banerjee, 2009*; *Sinenko et al., 2009*; *Mondal et al., 2011*; *Mukherjee et al., 2011*; *Tokusumi et al., 2011*; *Dragojlovic-Munther and Martinez-Agosto, 2012*; *Pennetier et al., 2012*; *Shim et al., 2012*; *Sinenko et al., 2012*). Wingless (Wg; Wnt in vertebrates) is expressed by blood progenitor cells in the lymph gland and has an important role in promoting their maintenance (*Sinenko et al., 2009*), and reactive oxygen species (ROS) function in these cells to potentiate blood progenitor differentiation both in the context of normal development and during oxidative stress (*Owusu-Ansah and Banerjee, 2009*). Progenitor cell maintenance at late developmental stages is also dependent upon Hedgehog (Hh) signaling from a small population of cells called the posterior signaling center that functions as a hematopoietic niche (PSC) (*Lebestky et al., 2003*; *Jung et al., 2005*).

More recently, it has been discovered that the maintenance of lymph gland blood progenitors also requires a backward signal arising from the differentiating cells (*Mondal et al., 2011*). This signal is controlled by a novel pathway that combines the function of the receptor tyrosine kinase Pvr and JAK-independent STAT (STAT92E) activation in differentiating cells, followed by the expression of ADGF-A (*Figure 1*), a secreted enzyme that converts adenosine to inosine (*Mondal et al., 2011*). Extracellular adenosine is a well-established signal in mammalian systems in various contexts, particularly stress conditions (*Fredholm, 2007*; *Sheth et al., 2014*), and an elevated adenosine level in *Drosophila* causes extensive blood cell proliferation (*Dolezal et al., 2005*; *Mondal et al., 2011*). It has been demonstrated that differentiating and mature cells express (and are the primary source of) ADGF-A, and that its enzymatic activity (which converts adenosine to inosine) is required for progenitor cell maintenance (*Mondal et al., 2011*). As differentiation proceeds, ADGF-A expression (activity) increasingly promotes the maintenance of extant blood progenitors through the reduction of stimulatory adenosine. In this way, the differentiating cell population helps balance the progenitor/differentiating cell ratio and is the basis for our referring to ADGF-A as an 'equilibrium signal'.

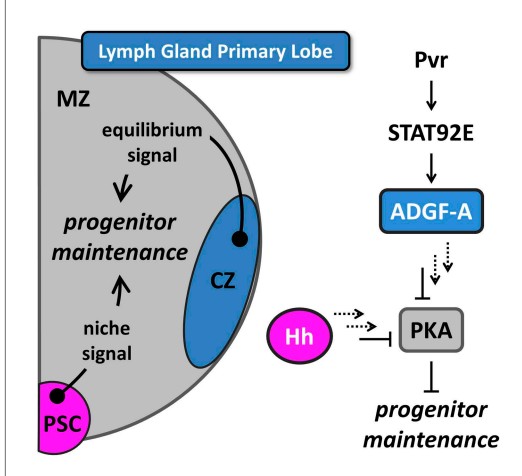

**Figure 1**. Equilibrium signaling maintains hematopoietic progenitors in the developing lymph gland. The lymph gland primary lobe consists of three distinct cellular populations or zones. The medullary zone (MZ) contains blood progenitor cells while the nearby cortical zone (CZ) contains differentiating and mature blood cells. The posterior signaling center (PSC) functions as a supportive population (a niche) that expresses Hedgehog (Hh) and maintains the progenitor cells utilizing this 'niche signal'. The receptor tyrosine kinase (RTK) Pvr and the STAT (STAT92E) transcriptional activator are required in CZ cells for the proper expression and secretion of the extracellular enzyme ADGF-A, which keeps the extracellular adenosine levels relatively low by converting it to inosine. The Pvr ligand Pvf1 is made in PSC cells and is transported through the lymph gland to activate Pvr in CZ cells. Collectively, we refer to the system that generates ADGF-A from the differentiating cells as 'equilibrium signaling', which is required independently of the niche-derived Hh signaling for the maintenance of progenitor blood cells in the MZ. Signaling events downstream of both ADGF-A and Hh (dashed arrows) cause the inhibition of Protein Kinase A (PKA) within progenitor blood cells, thereby promoting their maintenance. The individual components are color coded to match the schematic of the lymph gland. The equilibrium signal ADGF-A is blue, originating from the CZ; the niche signal Hh is magenta, originating in the PSC; PKA is gray, functioning in the MZ progenitor cells. Full details of this molecular pathway can be found in *Mondal et al., (2011)*.

Loss of ADGF-A (or STAT or Pvr) from differentiating cells increases extracellular adenosine level and thereby increases Adenosine Receptor (AdoR) signaling and downstream Protein Kinase A (PKA) activity in progenitors, which causes these cells to proliferate (*Dolezal et al., 2005*; *Mondal et al., 2011*). PKA is a central regulator of progenitor maintenance because it integrates input from both the equilibrium signal (ADGF-A) and the niche signal (Hh). PKA mediates the conversion of the transcriptional regulator Cubitus interruptus (Ci, a homolog of vertebrate Gli) from its full length form (Ci155), required for progenitor maintenance (*Mandal et al., 2007*), to a cleaved form (Ci75) that promotes proliferation. Signaling by Hh inhibits PKA and promotes Ci155 stabilization whereas adenosine/AdoR signaling activates PKA and promotes Ci75 conversion. Thus, both Hh (the niche signal) and ADGF-A (the equilibrium signal which removes adenosine) limit PKA activity and promote progenitor cell maintenance.

Although niche and equilibrium signaling are both clearly important, details of their regulation and interaction are less clear. Thus, we performed a loss-of-function genetic screen to identify new genes involved in lymph gland blood progenitor maintenance, particularly those involved in equilibrium signaling. In this study, we report the results of this screen and the identification of three genes, *bip1*, *RpS8*, and *Nup98*, as new components of the equilibrium signaling pathway.

## Results and discussion

### HHLT-gal4 and its use for whole-animal genetic screening during hematopoietic development

Unlike the adult eye or wing, analysis of internal larval tissues such as the lymph gland requires laborious dissection and processing. To circumvent this barrier to genetic screening, we generated a line of flies termed the *Hand-Hemolectin Lineage Traced-gal4* line (*HHLT-gal4 UAS-2XEGFP*, *Figure 2A*; see 'Materials and methods' for precise genotype) in which the hematopoietic system is labeled by Gal4-dependent expression of EGFP, such that it can be visualized in live, whole animals (*Figure 2B–C*). This line makes use of two

*gal4* drivers to target early lymph gland blood cells (hemocytes; *Hand-gal4*) and circulating and sessile blood cells (*Hemolectin-gal4* or *Hml-gal4*) and incorporates a Gal4/FLP recombinase-dependent cell lineage tracing cassette to maintain Gal4 expression in the lymph gland after the *Hand-gal4* driver itself is down-regulated during the first instar. The *Hand-gal4* driver is expressed in the embryonic cardiogenic mesoderm from which the lymph gland is derived. Therefore, the dorsal vessel (heart) cardioblasts and the pericardial nephrocytes are also marked by the cell lineage tracing cassette (*Figure 2B*). EGFP is not expressed in other larval tissues, except in the late third-instar salivary glands that are readily discernible from the hematopoietic system (*Figure 2B,E*).

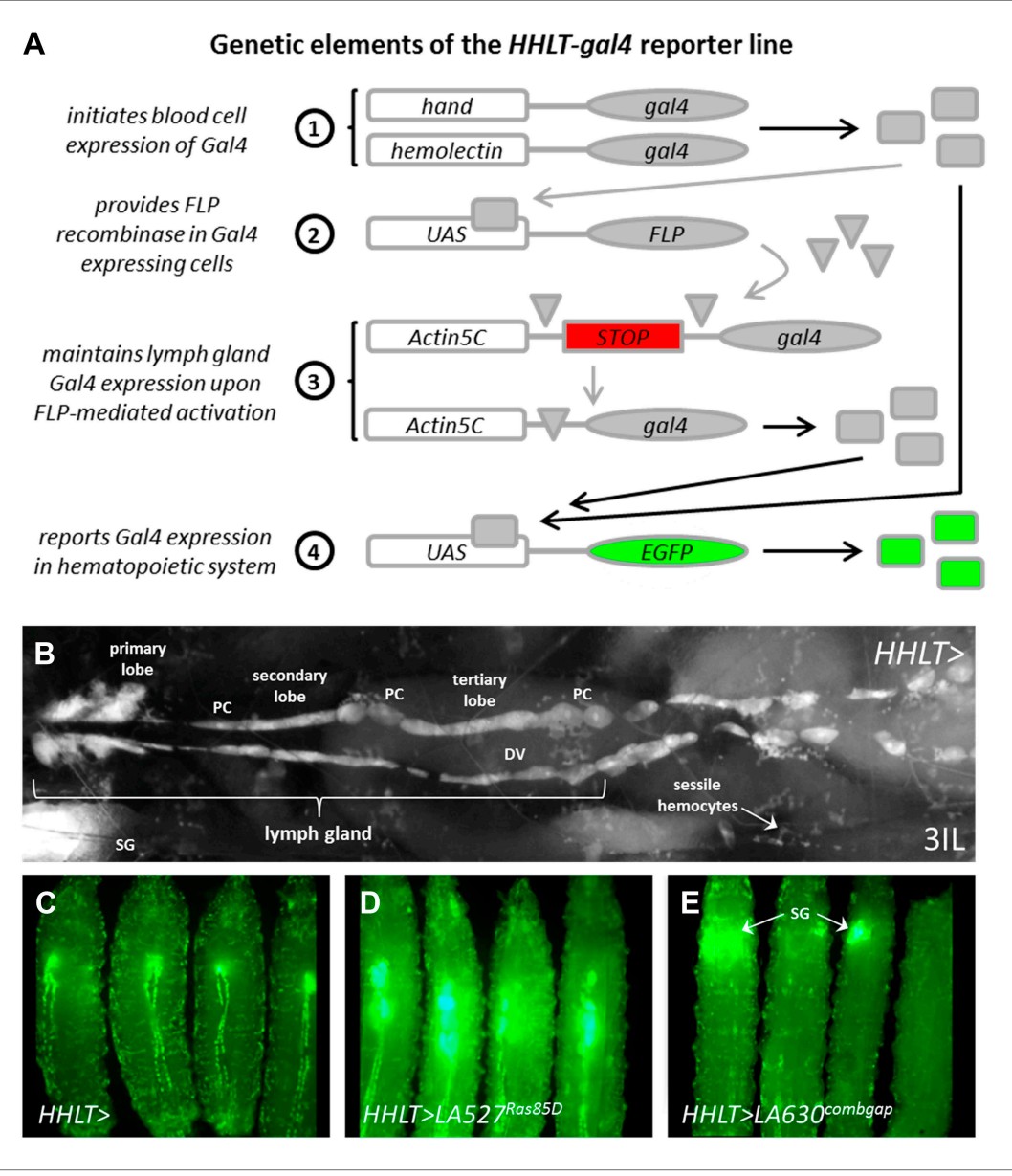

**Figure 2**. The *Hand-Hemolectin Lineage Tracing-gal4* line (*HHLT-gal4 UAS-2XEGFP*) and its use as an in vivo screening tool. (**A**) Schematic describing the key elements of the *HHLT-gal4* driver line. (**B**) Image showing the hematopoietic system within a wandering stage third-instar *HHLT > GFP* larva (dorsal view). Primary, secondary, and tertiary lobes of the lymph gland are readily discernible through overlying musculature, epidermal cells, and cuticle. Lymph gland lobes develop bilaterally, flanking the larval heart (dorsal vessel, DV). Non-blood pericardial cells (PC) also express GFP due to early expression of *Hand-gal4*. Circulating/sessile blood cells also express GFP due to *Hml-gal4* and sessile groups are easily observable. GFP is also seen in ventrally located salivary glands (SG, out of focus) of larvae beyond the third-instar transition (due to *Hand-gal4*). (**C**) *HHLT > GFP* control larvae; (**D**) *HHLT > GFP* larvae overexpressing *Ras85D* (*LA 527*) exhibit hyperproliferative lymph glands; (**E**) *HHLT > GFP* larvae overexpressing *combgap* (*LA 630*) show little or no GFP expression in the lymph gland region. Arrows indicate GFP fluorescence from salivary glands (SG).

The following figure supplement is available for figure 2:

**Figure supplement 1**. As a 'proof-of-principle' approach and to assess the effectiveness of *HHLT-gal4* as a screening tool, *HHLT-gal4* was crossed to lines harboring gain-of-function *UAS* transgenes known to cause excessive cellular proliferation, with the expectation that such transgenes would cause significant expansion of the hematopoietic tissues.

## Screening for progenitor maintenance genes in the developing lymph gland

A screen was conducted in which *HHLT-gal4* was used to independently misexpress 503 unique *UAS*-controlled *Drosophila* genes, with their effects on the hematopoietic system assessed in whole animals based upon EGFP expression. The particular collection of *UAS*-based gene misexpression lines used (termed LA lines) are mapped insertions (against Flybase release 5.7) of the *P{Mae-UAS.6.11}* element into endogenous gene loci that have been previously shown to cause developmental phenotypes upon misexpression (*Crisp and Merriam, 1997*; *Bellen et al., 2004*). When crossed to *HHLT-gal4*, 281 of these lines cause a scorable phenotype in either lymph glands or circulating blood cells of late third-instar larvae (*Supplementary file 1*). As an example, LA line 527, which is predicted to misexpress *Ras85D*, causes a robust expansion of the lymph gland (*Figure 2D*), consistent with the previously identified role of *Ras85D* in controlling hemocyte proliferation (*Asha et al., 2003*; *Sinenko and Mathey-Prevot, 2004*). By contrast, LA line 630 misexpresses the gene *combgap* (encoding a zinc finger transcription factor) and causes a strong reduction in lymph gland size (*Figure 2E*).

We used these results from the misexpression screen as a means to select potentially relevant genes for the subsequent loss-of-function analyses by RNA interference (using *UAS-RNAi* lines). We were able to obtain RNAi lines targeting 251 of the candidate genes identified by misexpression and found that 73 RNAi lines targeting 69 genes alter lymph gland size or morphology when crossed to *HHLT-gal4* (*Supplementary file 2*).

To characterize the RNAi phenotypes in more detail, the level of blood cell differentiation within the lymph gland was evaluated by immunostaining with anti-Peroxidasin (Pxn) antibodies (*Nelson et al., 1994*). In wild-type lymph glands, expression of mature cell markers such as Pxn is restricted to the periphery of the primary lobe (the cortical zone) (*Jung et al., 2005*). By contrast, when niche signaling or equilibrium signaling are compromised, progenitor cells are lost and differentiation markers, including Pxn, are expressed throughout the lymph gland primary lobes (*Mandal et al., 2007*; *Mondal et al., 2011*). Rescreening the 73 identified RNAi lines using *HHLT-gal4* identified 20 genes (21 RNAi lines) that, when knocked down, cause the expression of Pxn in cells throughout the lymph gland primary lobe (*Figure 3*; *Table 1* and *Supplementary file 2*). Compared to controls, the progenitor population (Pxn negative) is either strongly reduced or absent in each RNAi background. This 'expanded' Pxn phenotype is interpreted as a loss-of-progenitor cell phenotype.

## Zone-specific screening identifies putative equilibrium signaling genes

Using the pan-lymph gland *HHLT-gal4* driver, we identified 21 RNAi lines that cause a loss of progenitor cells in the primary lobes at late stages of lymph gland development. In order to discern whether any of the associated candidate genes have a specific progenitor-maintenance function that is restricted to cells belonging to a single zone, we rescreened the 21 RNAi lines using cell-type-specific Gal4-expressing lines. Targeting RNAi to differentiating and mature cells using *Hml-gal4* (*Sinenko and Mathey-Prevot, 2004*) identified six genes (CG6854 [*CTPsyn*], CG7398 [*Transportin*], CG7574 [*bip1*], CG10009 [*Noa36*], CG10198 [*Nup98*, also known as *Nup98-96*], and CG31938 [*Rrp40*]) that cause an expansion of Pxn (*Figure 4A–G*) and *Hml-gal4, UAS EGFP* (*Figure 4H–M*) expression. Since the function of these genes is needed in the CZ for the maintenance of the MZ progenitors, these six genes encode likely candidates for new components of the equilibrium signaling pathway.

Screening with *dome-gal4* (*Jung et al., 2005*) to target RNAi to the progenitor cells identified eleven genes (*Figure 4—figure supplement 1A–L*), three of which (*Transportin*, *Noa36*, and *Rrp40*) are in common with those identified using *Hml-gal4*. By contrast, use of *Antennapedia-gal4* (*Antp-gal4*) (*Mandal et al., 2007*) to target RNAi specifically to niche cells failed to identify any of the 21 lines as additional niche signaling components (not shown). Lastly, seven of the 21 RNAi lines did not cause a phenotype when expressed with any of the zone-specific Gal4 driver lines used. Taken together, our screen identified three genes, *CTPsyn*, *bip1*, and *Nup98*, which cause a loss of lymph gland progenitor cells upon RNAi knock down in differentiating cells, but not in progenitor cells or in niche cells. As described below, it was ultimately possible to connect two of these genes, *bip1* and *Nup98*, to the equilibrium signaling pathway through the control of Pvr expression.

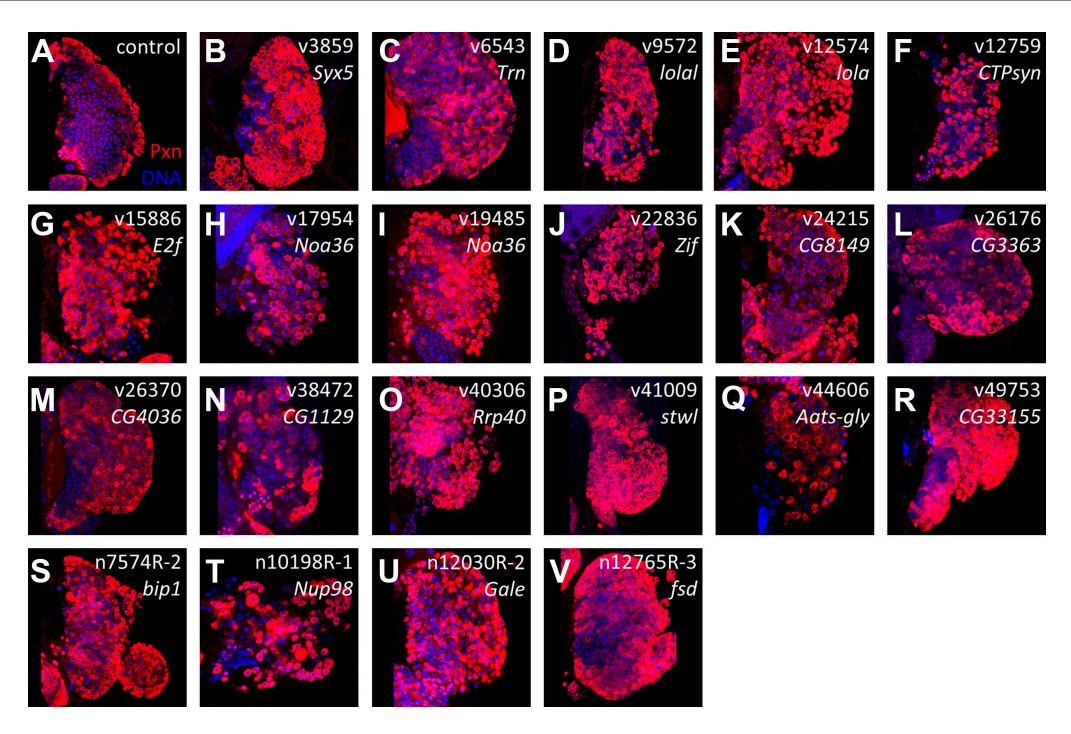

**Figure 3**. Identification of RNAi lines that cause an expanded Peroxidasin phenotype when expressed throughout the lymph gland. Peroxidasin (Pxn, *red*) is normally restricted to cortical zone cells (near the periphery) (**A**, control) but is seen throughout the lymph gland in RNAi backgrounds (**B**–**V**) expressed by *HHLT-gal4*. Line identifiers and gene targets are shown; additional details listed in *Table 1*. Images represent a single middle confocal section taken from a Z-plane series through the entire primary lobe.

## The bip1 gene functions in differentiating cells to regulate progenitor maintenance

The *bip1* gene was originally identified through a yeast two-hybrid screen that showed that its encoded protein binds the BTB/POZ domain of the transcription factor Bric à brac 1 (Bab1) (*Pointud et al., 2001*), a protein that has several developmental roles including the formation of ovarian terminal filament cells that are required for germline stem cell maintenance (*Lin and Spradling, 1993*; *Sahut-Barnola et al., 1995*; *Couderc et al., 2002*). Analysis of the predicted Bip1 amino acid sequence (InterPro) (*Hunter et al., 2012*) identifies a THAP domain containing a C2CH-type zinc finger motif that is known to bind DNA (*Sabogal et al., 2010*).

As no known *bip1* mutants exist, several different approaches were used to validate the *bip1* RNAi results and elucidate the function of the *bip1* gene in differentiating the blood cells. First, qRT-PCR confirmed that *bip1* is expressed in the lymph gland and demonstrated that the *bip1* RNAi line (NIG 7574R-2) actually targets *bip1* transcripts. Indeed, RNAi knock down of *bip1* using *Hml-gal4* (*Sinenko and Mathey-Prevot, 2004*; *Jung et al., 2005*) reduces *bip1* mRNA levels in the lymph gland to approximately ten percent of that observed in controls (*Figure 5A*). The *bip1* RNAi blood phenotype is also suppressible by the simultaneous overexpression of *bip1* (*UAS-bip1^LA645*; *Figure 5B–B′*), demonstrating the specific requirement for *bip1* in maintaining progenitors. Driving *bip1* RNAi with *Pxn-gal4*, an alternative differentiating- and mature-cell driver to *Hml-gal4*, also causes the loss of progenitor cells (*Figure 5C–C′*), thereby confirming that *bip1* knock-down in differentiating cells is key to its associated phenotype. Additionally, the progenitor cell marker *dome-MESO-lacZ* (*Hombria et al., 2005*; *Krzemien et al., 2007*) is strongly reduced relative to control lymph glands (*Figure 5D–D′*) in the *bip1* RNAi (*Hml-gal4*) background. This result confirms that progenitor cells fail to be maintained in *bip1* RNAi lymph glands, rather than ectopically upregulating Pxn and *Hml-gal4* expression.

**Table 1.** RNAi lines and target genes causing an 'expanded' Peroxidasin expression phenotype with *HHLT-gal4*

| Line # | UAS-RNAi ID | RNAi target | Gene | Off targets | LG size/quality | Protein function |
|---|---|---|---|---|---|---|
| 1 | 3859 | CG4214 | *Syx5* | 0 | Small/missing | Golgi SNARE |
| 2 | 6543 | CG7398 | *Trn* | 1 | Large/baggy | hnRNP nuclear import |
| 3 | 9572 | CG5738 | *lolal* | 0 | Small | Transcription factor |
| 4 | 12574 | CG12052 | *lola* | 0 | Large/baggy | Transcription factor |
| 5 | 12759 | CG6854 | *CTPsyn* | 0 | Small/baggy | CTP synthase |
| 6 | 15886 | CG6376 | *E2f* | 1 | Small/normal | Transcription factor |
| 7 | 17954 | CG10009 | *Noa36* | 0 | Small/missing | Zinc finger nucleolar protein |
| 8 | 19485 | CG10009 | *Noa36* | 0 | Small/missing | Zinc finger nucleolar protein |
| 9 | 22836 | CG10267 | *Zif* | 0 | Small/missing | Transcription factor |
| 10 | 24215 | CG8149 | *CG8149* | 0 | Baggy | *DNA binding protein* |
| 11 | 26176 | CG3363 | *CG3363* | 0 | Small | Unknown |
| 12 | 26370 | CG4036 | *CG4036* | 1 | Large | *Oxidoreductase* |
| 13 | 38472 | CG1129 | *CG1129* | 0 | Small/missing | *Peptide transferase* |
| 14 | 40306 | CG31938 | *Rrp40* | 0 | Small/normal | RNA exosome |
| 15 | 41009 | CG3836 | *stwl* | 0 | Small/normal | Transcription factor |
| 16 | 44606 | CG6778 | *Aats-gly* | 0 | Small/missing | Glycyl-tRNA synthetase |
| 17 | 49753* | CG33155 | *CG33155* | 4 | Small/normal | Unknown |
| 18 | 7574R-2 | CG7574 | *bip1* | 0 | Small/baggy | *Transcription factor* |
| 19 | 10198R-1 | CG10198 | *Nup98-96* | 0 | Small/missing | Nucleoporin |
| 20 | 12030R-2 | CG12030 | *Gale* | 0 | Small | UDP-galactose 4'-epimerase |
| 21 | 12765R-3 | CG12765 | *fsd* | 0 | Small/normal | F-box protein |

*This RNAi line targeting sequence overlaps with the putative *mRpL53* gene in the same locus. Lines 1–17 from VDRC, lines 18–21 from NIG Japan.

Several ribosomal components, including Ribosomal protein S8 (RpS8), have been shown to associate with chromatin at active transcription sites and to associate with nascent transcripts to form ribonucleoprotein complexes (*Brogna et al., 2002*). Interestingly, RpS8 has also been identified in genomic-scale yeast two-hybrid analyses as a Bip1-interacting protein (*Giot et al., 2003*; *Formstecher et al., 2005*; *Stark et al., 2006*), which suggests that Bip1 and RpS8 may function together in vivo to regulate gene expression. Consistent with this idea, RNAi knockdown of *RpS8* also causes the expanded expression of both Pxn and *Hml-gal4 UAS-GFP* expression throughout the lymph gland primary lobes (*Figure 5E–E'*). This result reflects a specific function of RpS8 in these cells because knockdown directly in niche or progenitor cells (using *Antp-gal4* and *dome-gal4*, respectively) does not cause their loss to differentiation (not shown). Thus, *RpS8* RNAi effectively phenocopies *bip1* RNAi, as both cause a loss of progenitor cells when knocked down in differentiating cells. Collectively, these data support a model in which Bip1 functions along with RpS8 in a protein complex within differentiating cells to maintain multipotent lymph gland progenitors at later stages of development, consistent with a potential function in the equilibrium signaling pathway.

## bip1 functions genetically upstream of the equilibrium signaling pathway

The progenitor maintenance function of Pvr signaling in differentiated cells requires the downstream function of the STAT transcriptional activator and the secreted enzyme ADGF-A (*Mondal et al., 2011*),

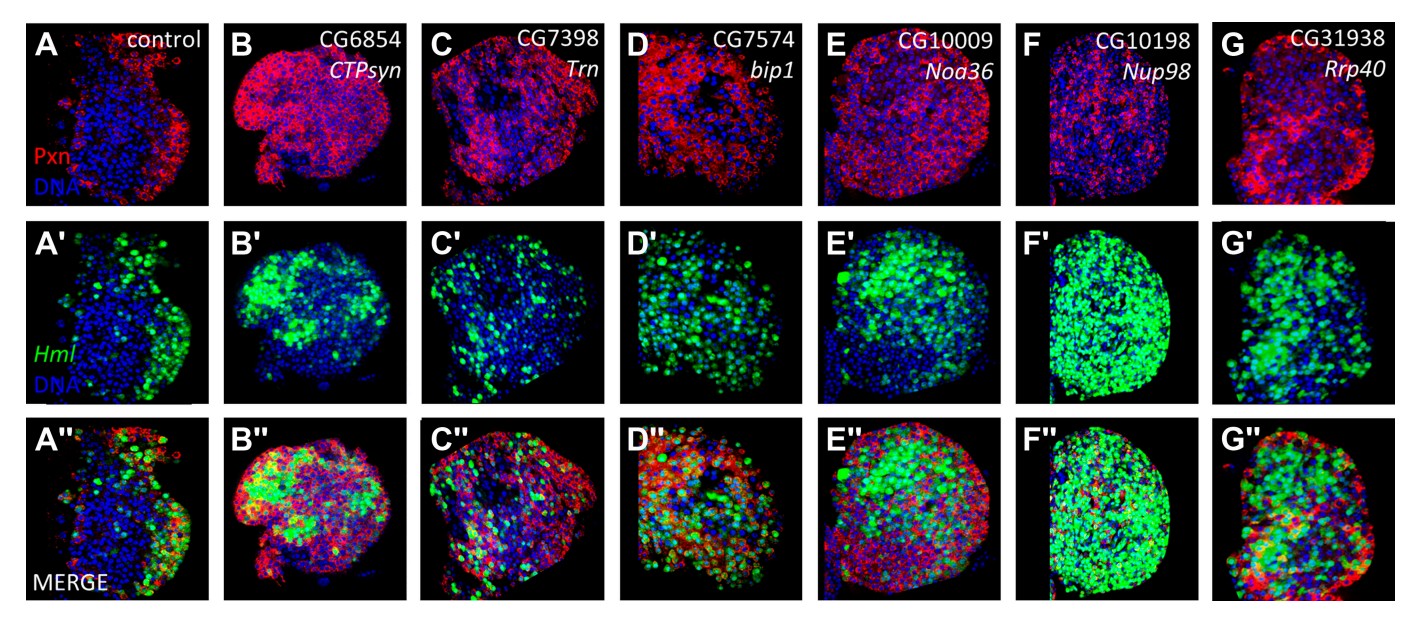

**Figure 4**. Identification of candidate genes that cause an expanded Peroxidasin expression phenotype within the lymph gland when knocked down by RNAi in differentiating and mature cells. RNAi from identified lines (*Figure 3*/*Table 1*) was expressed in lymph glands using *Hml-gal4 UAS-GFP* (*Hml > GFP*). In the control, Pxn (**A**) and GFP (**A′**) are restricted to the cortical zone (periphery). By contrast, knock down of six candidate genes causes extensive expression of Pxn (**B–G**) and *Hml* (*Hml > GFP*) (**B′–G′**) throughout the lymph gland, indicating a loss of progenitors in these genetic backgrounds. The combined Pxn and *Hml* expression patterns for each genetic background are shown (MERGE, **A″–G″**). DNA (blue) is stained to mark nuclei.

The following figure supplement is available for figure 4:

**Figure supplement 1**. RNAi lines causing an expanded Peroxidasin expression phenotype when expressed in progenitor cells using *dome-gal4*.

and, consistent with this relationship, overexpression of either activated STAT (STAT[ACT]) (*Ekas et al., 2010*) or ADGF-A in differentiating cells can suppress the *Pvr* loss-of-function phenotype (*Mondal et al., 2011*). Likewise, we find that overexpression of STAT[ACT] or ADGF-A can suppress the *bip1* RNAi phenotype (*Figure 6A–D′*). Furthermore, overexpression of *Pvr* also strongly suppresses the *bip1* RNAi phenotype, returning lymph gland morphology and organization to essentially wild type (*Figure 6E–E′*). By contrast, overexpression of *bip1* does not suppress the *Pvr* RNAi phenotype (*Hml-gal4 UAS-Pvr RNAi UAS-bip1[LA645]*; not shown). Collectively, these results place *bip1* function genetically upstream of *Pvr* and other equilibrium signaling components in lymph gland progenitor maintenance by differentiating cells.

## Bip1 and RpS8 control the equilibrium signaling pathway by regulating Pvr expression

The suppression of the *bip1* RNAi phenotype by overexpression of *Pvr* suggested that *bip1* may positively control *Pvr* expression during normal development. Indeed, a reduction in Pvr protein expression within the lymph gland is observed in the *bip1* RNAi background by mid-second instar, soon after differentiation begins (~40 hr post-hatching; *Figure 6F–J′*). This reduction in Pvr expression is even stronger (along with significantly increased differentiation, based upon *Hml-gal4* expression) at the same developmental time point in larvae having two copies of *Hml-gal4 UAS-bip1 RNAi* (compare *Figure 6I′* with *Figure 6H′*), further supporting the model that *bip1* RNAi causes the loss of Pvr expression. By the late third instar (when the *bip1* RNAi differentiation phenotype is most apparent), Pvr protein levels in the lymph gland remain strongly reduced (*Figure 6J–J′*).

Knockdown of *bip1* function in lymph glands by *Hml-gal4*-mediated RNAi reduces lymph gland *Pvr* transcript levels to approximately 70% of control levels (assessed by qRT-PCR; *Figure 6—figure supplement 1A*), consistent with the observed loss of Pvr protein (*Figure 6H′–J′*). However, because not all lymph gland cells express *Hml* (*Hml-gal4*), the actual reduction of *Pvr* transcript levels in cells expressing *bip1* RNAi is likely to be greater than the observed total reduction. In support of this idea,

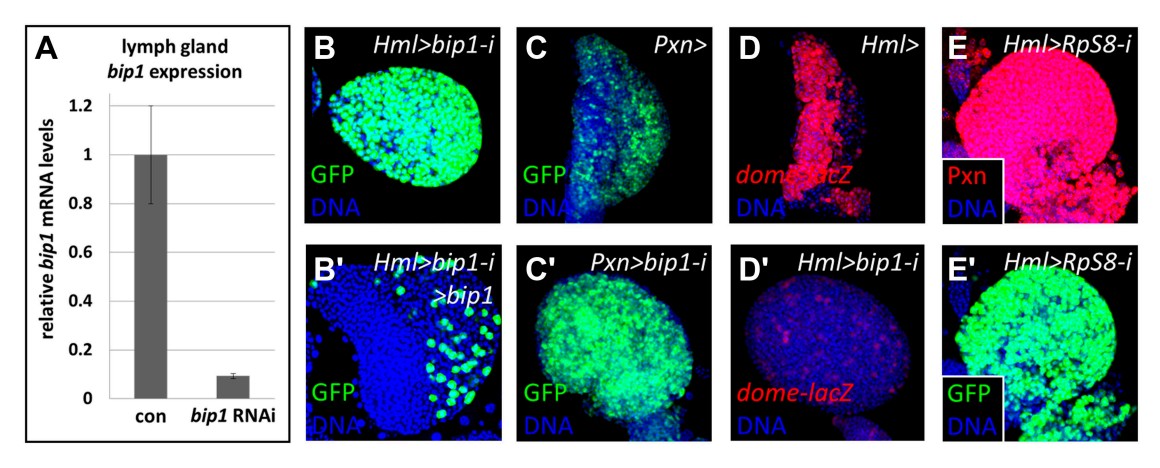

**Figure 5.** Validation of the *bip1* RNAi phenotype. (**A**) Quantitative RT-PCR demonstrates that *bip1* is expressed in the lymph gland and that the RNAi line NIG 7574R-1 targeting *bip1* indeed reduces *bip1* transcript level when expressed using *Hml-gal4*. *Hml-gal4* expresses GFP throughout the primary lobes in *bip1* RNAi lymph glands (*Hml > bip1-i*, **B**), and this phenotype is suppressed by overexpression of *bip1* (**B'**), restoring both the cortical and the medullary zones. (**C–C'**) Expression of *bip1* RNAi using *Pxn-gal4* phenocopies obtained with *Hml-gal4*, further supporting a cell-type-specific function of *bip1*. Expression of the progenitor cell marker *dome-MESO-lacZ* (**D**) is strongly reduced in *bip1* RNAi lymph glands (**D'**), demonstrating that the gain in differentiation markers is due to the loss of progenitor cells that normally express *dome-MESO-lacZ*. RNAi knock down of *RpS8*, encoding a putative Bip1-interacting protein, causes the expansion of Pxn and *Hml-gal4* expression throughout the lymph gland (**E–E'**), similar to that observed upon the loss of *bip1*.

*bip1* RNAi in circulating blood cells (where greater than 90% express *Hml-gal4*) reduces *Pvr* transcript level to approximately 35% of the control level (***Figure 6—figure supplement 1B***). Expression of *bip1* RNAi in FLP-out Gal4-expressing cell clones made exclusively in the lymph gland strongly reduces Pvr levels compared to nearby cells not expressing RNAi as well as to mock clones (***Figure 6—figure supplement 2A–C***), consistent with the autonomous regulation of Pvr by *bip1* in the lymph gland. Furthermore, *bip1* RNAi expression with *srpHemo-gal4* (***Bruckner et al., 2004***), which expresses in a large fraction of circulating cells but in few or no cells within the lymph gland, does not reduce lymph gland Pvr levels (***Figure 6—figure supplement 2E–H***). Collectively, these data indicate that *bip1* is required for proper Pvr protein expression, and therefore proper equilibrium signaling, within the developing lymph gland.

As described above, RpS8 is a putative Bip1-interacting protein in vivo and *RpS8* RNAi in differentiating lymph gland cells, like *bip1* RNAi, causes the loss of progenitor cells (***Figure 5F–F'***). This effect is likely due to the loss of equilibrium signaling during development since *RpS8* RNAi also reduces Pvr protein expression in the lymph gland (***Figure 6K–K'***). Knockdown of *RpS8* by RNAi, as with knockdown of *bip1*, also reduces *Pvr* transcript levels (***Figure 6—figure supplement 1C***). Interestingly, a *Drosophila* RNAi screen using the blood-related S2 cell line previously identified both Pvr and RpS8 as regulators of cell size and division (***Sims et al., 2009***). Although the relationship between Pvr and RpS8 was not explored, their results as well as ours are consistent with RpS8 having a regulatory role in *Pvr* expression in blood cells.

## Nup98 also regulates Pvr expression

In addition to *bip1*, the screen described here identified *Nup98* as a potential equilibrium signaling component because its knockdown in differentiating cells specifically causes a loss of progenitors cells (***Figure 3T*** and ***Figure 4F–F''***). Although Nup98 is widely known as a general component of the nuclear pore complex, recent work has demonstrated that Nup98 and other nuclear pore components such as Sec13 and Nup88, can regulate gene expression through the binding of target promoters (***Capelson et al., 2010***; ***Kalverda et al., 2010***; ***Liang et al., 2013***). Moreover, chromatin immunoprecipitation experiments identified *bip1*, *RpS8*, and the equilibrium signaling genes *Pvr* and *STAT* (*STAT92E*) as in vivo Nup98 regulatory targets (***Capelson et al., 2010***). Consistent with a function in regulation of equilibrium signaling genes, *Nup98* knockdown specifically in differentiating cells of lymph glands causes a strong reduction in Pvr expression (***Figure 6L–L'***). By contrast, RNAi knockdown

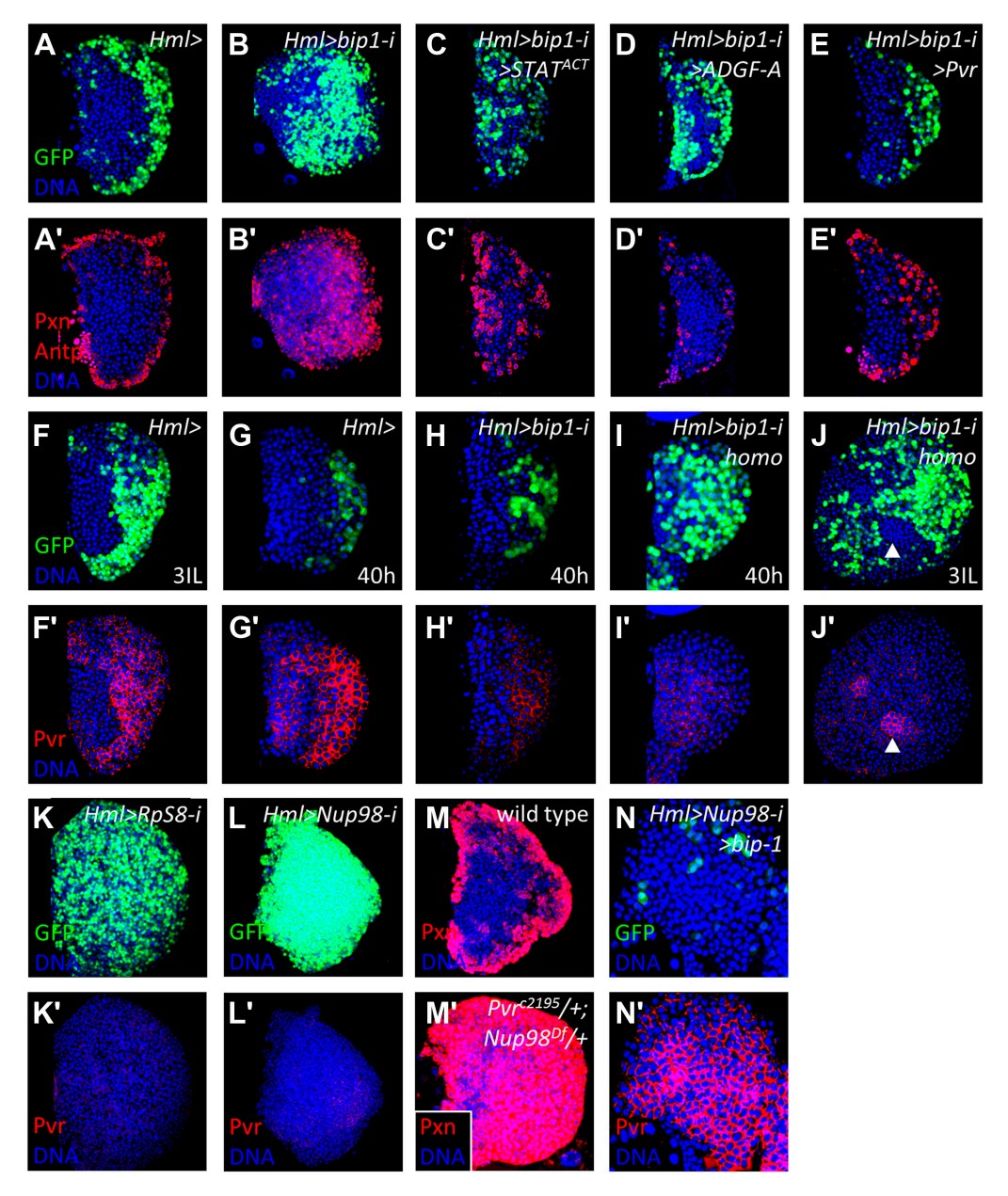

**Figure 6**. *bip1*, *RpS8*, and *Nup98* control Pvr expression in the lymph gland. Expression of *bip1* RNAi in differentiating cells (*Hml-gal4 or Hml>*, **B–B'**) causes expansion of both Pxn and *Hml-gal4 UAS-GFP* throughout the lymph gland, as compared to controls (**A–A'**). Misexpression of either activated STAT (STAT$^{ACT}$, **C–C'**), ADGF-A (**D–D'**), or Pvr (**E–E'**) partially (in the case of STAT activation or ADGF-A overexpression) or fully (in case of Pvr overexpression) suppresses this *bip1* phenotype, suggesting that *bip1* functions upstream of these genes. Expression of Pvr in control third-instar lymph glands (**F–F'**) and mid-second instar (40 hr post-hatching, **G–G'**). Reduced expression of Pvr is already apparent in *bip1* RNAi lymph glands by 40 hr (**H–H'**), and this loss is even stronger in homozygous animals expressing higher levels of RNAi (**I–I'**); increased differentiation, based upon *Hml-gal4 UAS-GFP* expression, is also apparent (**I**). Strong suppression of Pvr is also observed in homozygous *bip1* RNAi lymph glands (**J–J'**). RNAi knockdown of *RpS8* also causes differentiation and the loss of Pvr expression (**K–K'**). Likewise, RNAi knockdown of *Nup98* also causes differentiation and the loss of Pvr expression (**L–L'**). (**M**) Control background (*Hml-gal4/+*) showing normal expression of the differentiation marker Pxn in the cortical zone of the lymph gland. Progenitor cells in the MZ region are easily discerned by their lack of Pxn expression. By contrast, few progenitor cells (Pxn-negative cells) are observed in lymph glands when single-copy loss-of-function mutations of
*Figure 6. Continued on next page*

*Figure 6. Continued*

*Pvr* and *Nup98* (*Pvr*$^{C2195}$/+; *Nup98*$^{Df(3R)mbc-R1}$/+) are combined (**M'**), further indicating the close interaction between these genes. The middle-third (confocal z-stack) of the primary lobe is shown. Misexpression of *bip1* in this background is sufficient to suppress these phenotypes and restore Pvr expression to the lymph gland (**N–N'**).

The following figure supplements are available for figure 6:

**Figure supplement 1**. *bip1* and *RpS8* are required for normal *Pvr* transcript levels.

**Figure supplement 2**. Pvr expression is regulated autonomously by *bip1*, *Nup98*, and *RpS8* within the lymph gland.

**Figure supplement 3**. Loss of the nucleoporin Sec13 by RNAi neither causes a differentiation phenotype within the lymph gland nor the loss of Pvr expression.

**Figure supplement 4**. Loss of Pvr expression is not a common feature of highly differentiated lymph glands.

**Figure supplement 5**. Overexpression of *bip1*, *Nup98*, and *RpS8*, and RNAi knockdown of other nucleoporins does not affect Pvr levels in the lymph gland.

---

of the nucleoporin Sec13 in differentiating cells has no effect on the maintenance of progenitor cells or Pvr expression (*Figure 6—figure supplement 3*) underscoring the specific role of *Nup98* in Pvr expression control. Furthermore, the close genetic relationship between *Nup98* and *Pvr* is illustrated by the fact that single-copy loss of these genes in combination causes extensive loss of progenitor cells to differentiation (*Figure 6M–M'*). Interestingly, overexpression of *bip1* in *Nup98* RNAi lymph glands (*Hml-gal4 UAS-Nup98 RNAi UAS-bip1*$^{LA645}$) is sufficient to restore Pvr protein expression and to suppress the loss of progenitors to differentiation (based upon lymph gland morphology and *Hml-gal4* expression; *Figure 6N–N'*).

As has been shown, knockdown of *bip1*, *Nup98*, or *RpS8* in differentiating cells each causes a strong reduction in Pvr expression in the lymph gland. Our interpretation of this common phenotype is that each gene works in the equilibrium signaling pathway to control Pvr expression, although an alternative hypothesis is that the loss of Pvr expression is a common feature of highly differentiated lymph glands and is not specifically related to the function of these genes. To test this, Pvr expression was examined in *collier* (*col*) mutant lymph glands, which lack niche signaling and are strongly differentiated by late larval stages (*Crozatier et al., 2004*; *Mandal et al., 2007*), and was found to be normal (*Figure 6—figure supplement 4*, compare with Pvr expression in wild-type cortical zone differentiating cells in *Figure 6F'*). Thus, *Pvr* requires *bip1*, *RpS8*, and *Nup98* for proper developmental expression in the lymph gland.

Several genetic screens, including overexpression and enhancer/suppressor screens of mutant or tumor phenotypes, have been conducted in the fly hematopoietic system (*Milchanowski et al., 2004*; *Zettervall et al., 2004*; *Stofanko et al., 2008*; *Avet-Rochex et al., 2010*; *Tan et al., 2012*; *Tokusumi et al., 2012*); however, the screen described here represents the first loss-of-function screen targeting normal developmental mechanisms throughout the lymph gland. This was accomplished with the development and use of the pan-lymph gland expression tool *HHLT-gal4* to drive *UAS*-mediated RNAi, which identified 20 different candidate genes that cause a loss of progenitor cells when knocked down within the lymph gland. From subsequent analyses using lymph gland zone-restricted Gal4 driver lines, we arrive at a model (*Figure 7*) in which Bip1, RpS8, and Nup98 are required in differentiating blood cells upstream of Pvr to control its expression and function in the equilibrium signaling pathway that maintains blood progenitors within the lymph gland. Future analyses will be required to identify additional components of this important signaling pathway and to provide more information about how equilibrium signaling interacts with other pathways in the control of blood cell progenitor maintenance, cell fate specification, and proliferation.

The Pvr receptor, with its numerous developmental roles, is arguably one of the most important members of the *Drosophila* RTK family, yet most of what is known about Pvr stems from analyses of how it works in the context of intracellular signaling. Little is known about how *Pvr* gene or protein

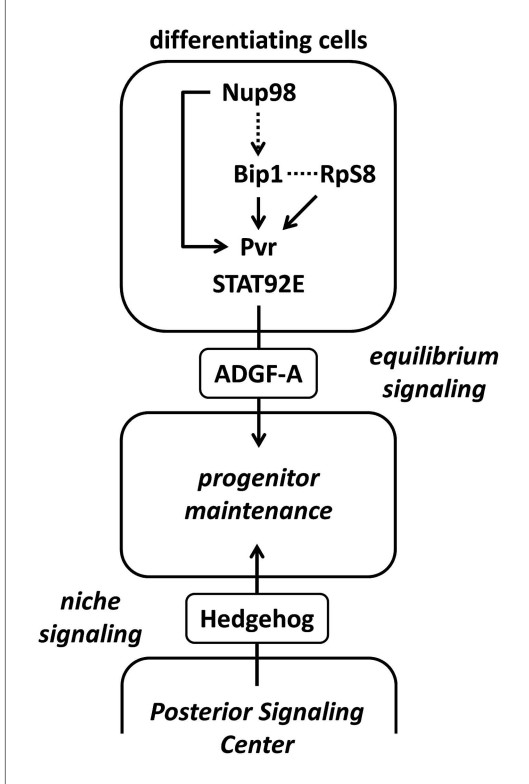

**Figure 7**. Schematic of the equilibrium signaling pathway demonstrating the proposed roles of Bip1, RpS8, and Nup98 in controlling Pvr. Bip1, RpS8, and Nup98 are independently required for the expression of Pvr (direct arrows). Rescue of endogenous Pvr expression by misexpression of *bip1* in the *Nup98* RNAi background indicates that *bip1* functions genetically downstream of *Nup98* (dashed arrow) in the control of Pvr expression. Bip1 and RpS8 may work together in a complex (dashed line) to control Pvr expression in vivo. These components collectively comprise the known equilibrium signaling pathway working within the lymph gland to promote progenitor cell maintenance, along with the previously known Hh niche signaling mechanism.

expression is regulated. Importantly, the work described here sheds new light upon this issue by demonstrating a role for *bip1*, *RpS8*, and *Nup98* in the regulation of *Pvr* expression. Our data and that of others suggest that this regulation of *Pvr* is likely taking place at the gene level, although other mechanisms are also possible. Ribosomes are required for protein translation, however specific ribosomal components or subunits may selectively stabilize transcripts and/or mediate preferential translation (*Xue and Barna, 2012*), while nucleoporins control both nuclear entry of regulatory proteins and the exit of mRNAs to the cytoplasm, and specific subcomponents are known to exhibit differential functions in this regard (*Strambio-De-Castillia et al., 2010*). Thus, RpS8 and Nup98 may selectively affect *Pvr* expression post-transcriptionally through transcript stabilization, transport, and translation. Although the specific mechanisms of molecular control of *Pvr* expression by *bip1*, *RpS8*, and *Nup98* remain to be determined, their function is clearly critical in mediating proper equilibrium signaling and, therefore, proper blood progenitor maintenance within the lymph gland. The finding that *bip1* regulates Pvr expression in the context of hematopoietic equilibrium signaling represents the first functional association for *bip1* in *Drosophila*. The predicted Bip1 protein exhibits only one recognizable structural sequence, namely a THAP domain that contains a putative DNA-binding zinc finger motif. Our results suggest that Bip1 behaves as a positive regulator of *Pvr* transcription, but whether this occurs directly through Bip1 interaction with the *Pvr* locus will require further investigation.

Understanding how progenitor cell maintenance and homeostasis is controlled over developmental time is crucial for understanding normal cellular and tissue dynamics, especially in the context of ageing or disease. The identification of Bip1 and Nup98 as regulators of hematopoietic progenitors in *Drosophila* may be indicative of important conserved functions of related proteins within the vertebrate blood lineages similar to what has been shown previously for GATA, FOG, and RUNX factors (*Waltzer et al., 2010*). THAP-domain proteins are conserved across species and have been reported to have a variety of important functions in mammalian systems, including maintenance of murine embryonic stem cell pluripotency (*Cayrol et al., 2007*; *Dejosez et al., 2008*, *2010*). What role, if any, THAP-domain proteins have in vertebrate blood progenitor maintenance (or hematopoiesis in general) remains to be established. Likewise, Nup98 has not been implicated in any normal hematopoietic role despite being a well-studied protein in other contexts.

With regard to the diseased state, mutations in the human *THAP1* gene have been associated with dystonia (*Fuchs et al., 2009*; *Paisan-Ruiz et al., 2009*; *Kaiser et al., 2010*; *Mazars et al., 2010*), a neuromuscular disorder that causes repetitive, involuntary muscular contraction, and THAP1/Par4 protein complexes have been shown to promote apoptosis in leukemic blood cells in various experimental contexts in vitro (*Lu et al., 2013*; *Zhang et al., 2014*). Chromosomal translocations that generate Nup98 fusion proteins have been implicated in numerous human myelodysplastic

syndromes and leukemias (*Nishiyama et al., 1999*; *Ahuja et al., 2001*; *Lin et al., 2005*; *Nakamura, 2005*; *van Zutven et al., 2006*; *Slape et al., 2008*; *Kaltenbach et al., 2010*; *Murayama et al., 2013*), further underscoring the need to explore Nup98 function in the hematopoietic system. Therefore, the study of *bip1* and *Nup98* in *Drosophila*, a powerful molecular genetic system, will likely be of benefit to understand the function of related vertebrate genes in normal and disease contexts.

## Materials and methods

### Fly stocks

Misexpression *P{Mae-UAS.6.11}* inserts (LA lines) were obtained from John Merriam, UCLA (Los Angeles, California). *UAS-RNAi* lines were obtained from the Vienna *Drosophila* RNAi Center (VDRC, Vienna, Austria), the National Institute of Genetics (NIG, Kyoto, Japan), and the Bloomington *Drosophila* Stock Center (TRiP lines, BDSC, Bloomington, Indiana). The lines *UAS-FLP.JD1*, *UAS-2XEGFP*, *P{GAL4-Act5C(FRT.CD2).P}S*, *UAS-human Raf^{ACT}*, *Df(3R)mbc-R1*, *UAS-RpS8^{PD01446}*, and *w^{1118}* (BDSC 5905) were from the BDSC. *Pvr^{c02195}* was from Exelixis (available from BDSC, obtained from D Montell). *Hml^{Δ}-gal4 UAS-2XEGFP* (S Sinenko), *Antp-gal4/TM6B Tb* (S Cohen), *P{ubi-gal80 ts}10; Antp-gal4/TM6B Tb* (this lab), *domeless-gal4 UAS-2XEYFP/FM7i* (this lab), *UAS-DAlk^{ACT}* (R Palmer), *dome-MESO-lacZ* (S Brown), *Pxn-gal4* (M Galko), *UAS-STAT^{ACT}* (E Bach), *UAS-ADGF-A* (T Dolezal), *collier^{1}; P(col5-cDNA)/CyO-TM6B, Tb* (M Crozatier), *srpHemo-gal4* (K Brückner), and *Hand-gal4* (Z Han) have been previously described.

### HHLT-gal4 construction and whole animal screening

Second chromosome inserts of *Hand-gal4*, *Hml∆-gal4*, *UAS-FLP.JD1*, and *UAS-2XEGFP* were recombined onto a single chromosome and placed with *P{GAL4-Act5C(FRT.CD2).P}S* on Chromosome 3. Because Gal4 reporter lines with specific, pan-lymph gland expression are unknown, we took advantage of a FLP-out lineage tracing approach that we have used previously to perpetually mark lymph gland cells (*Jung et al., 2005*; *Evans et al., 2009*). The *Hand-gal4* reporter reflects the expression of the *Hand* gene, which is expressed in the cardiogenic mesoderm, from which the lymph gland is derived. Within the lymph gland, *Hand-gal4* is expressed from the late embryo through the first larval instar but then is downregulated (*Han and Olson, 2005*). Using *Hand-gal4* in conjunction with *UAS-FLP* and a FLP-out Gal4-expressing line (*P{GAL4-Act5C(FRT.CD2).P}S*) (*Pignoni and Zipursky, 1997*), lymph gland cells are perpetually with EGFP throughout all subsequent developmental stages. To express EGFP in circulating cells, we used *Hemolectin-gal4* (*Hml∆-gal4*) (*Sinenko and Mathey-Prevot, 2004*), which is specific to mature blood cells both in circulation and in the lymph gland cortical zone (*Jung et al., 2005*). *HHLT-gal4* expression is easily detectable in lymph glands and circulating cells of whole animals throughout larval development. Due to the embryonic activity of *Hand-gal4*, *HHLT-gal4* also labels dorsal vessel cardioblasts and pericardial cells, although by late larval stages the expression of EGFP in the former is almost undetectable.

 *HHLT-gal4* virgins were crossed to males from individual LA lines, RNAi lines, or *w^{1118}* as a control. All crosses were reared at 29°C to maximize Gal4 activity. Wandering third-instar larvae from control and experimental crosses were collected, washed with water, and placed in glass spot wells (Fisher) on ice to minimize movement. Animals were scored visually using a Zeiss Axioskop 2 compound fluorescence microscope. Non-screen images of *HHLT > GFP* larvae were collected with a Zeiss SteREO Lumar fluorescence microscope. Images were collected using either an AxioCam HRc or HRm camera with AxioVision software.

### Tissue dissection and antibody staining and analysis

Lymph glands were dissected and processed as previously described (*Jung et al., 2005*). Briefly, lymph glands were dissecting from third-instar larvae in 1× PBS, fixed in 4% formaldehyde/1× PBS for 30 min, washed three times in 1×PBS with 0.4% Triton-X (1× PBST) for 15 min each, blocked in 10% normal goat serum/1× PBST for 30 min, followed by incubation with primary antibodies in block. Primary antibodies were incubated with tissue overnight at 4°C and then washed three times in 1× PBST for 15 min each, reblocked for 15 min, followed by incubation with secondary antibodies for 3 hr at room temperature. Samples were washed three times in 1× PBST, with TO-PRO-3 iodide (diluted 1:1000; Invitrogen, Carlsbad, California) added to the last wash to stain nuclei. Samples were washed briefly with 1× PBS to remove excess TO-PRO-3 and detergent prior to mounting on glass slides in VectaShield (Vector Laboratories, Burlingame, California). Mouse anti-Peroxidasin was a kind gift from John and

Lisa Fessler (UCLA) and was used at 1:1500 dilution. Rat anti-Pvr was a kind gift from Benny Shilo and was used at 1:400 dilution. Secondary Cy3-labeled antibodies were obtained from Jackson ImmunoResearch Laboratories Inc. (West Grove, Pennsylvania) and used at 1:500 dilution.

## Quantitative real-time PCR analysis

Lymph glands from 50 third-instar larvae were isolated by dissection. For fat body analysis, ten third-instar larvae were used. RNA was extracted from these tissues with the RNeasy mini kit (Qiagen, Germantown, Maryland). Relative quantitative RT-PCR (comparative CT) was performed using Power SYBR Green RNA-to-CT 1-step kit (Applied Biosystems, Carlsbad, California) and a StepOne Real-Time PCR detection thermal cycler (Applied Biosystems) using primers specific for *Pvr*, *bip1*, and *rp49*. Primer sequences are: *Pvr*(forward), 5'-TTCGGATTTCGATGGTGAAT-3'; *Pvr*(reverse), 5'-CGGACACTAAGCTGGTCGAT-3'; *bip1*(forward), 5'-CGGAGTTTATGGACAGCACA-3'; *bip1*(reverse), 5'-CCTTAGCAGGAGGAGGAGGT-3'; *rp49*(forward), 5'-GCTAAGCTGTCGCACAAATG-3'; *rp49*(reverse), 5'-GTTCGATCCGTAACCGATGT-3'.

## Acknowledgements

We thank Amir Yavari, Julia Manasson, Tanya Hioe, Sean Mofidi, Jesse Zaretsky, and Tina Muhkerjee for assistance generating the *HHLT-gal4* line and with various aspects of the screen. We thank John Merriam (UCLA) for use of the LA lines and John and Lisa Fessler (UCLA) for the kind gift of anti-Pxn antibodies. Lastly, we thank Ira Clark, John Merriam, and members of the Banerjee Lab for discussion of the project and comments on the manuscript.

## Additional information

### Competing interests

UB: Reviewing editor, *eLife*. The other authors declare that no competing interests exist.

### Funding

| Funder | Grant reference number | Author |
|---|---|---|
| National Heart, Lung, and Blood Institute | R01 HL067395 | Utpal Banerjee |
| California Institute for Regenerative Medicine | TG2-01169 | Jiwon Shim |

The funders had no role in study design, data collection and interpretation, or the decision to submit the work for publication.

### Author contributions

BCM, JS, Acquisition of data, Analysis and interpretation of data; CJE, Conception and design, Acquisition of data, Analysis and interpretation of data, Drafting or revising the article; UB, Conception and design, Analysis and interpretation of data, Drafting or revising the article

### Author ORCIDs

Jiwon Shim, http://orcid.org/0000-0003-2409-1130

## Additional files

### Supplementary files

• Supplementary file 1. Genes and phenotypes associated with *P{Mae-UAS.6.11}* LA insertion lines expressed with *HHLT-gal4*. Screen and line identifiers are shown along with the predicted misexpressed gene and the associated whole-animal lymph gland and circulating cell phenotype.

• Supplementary file 2. Genes and phenotypes associated with *UAS-RNAi* lines expressed with *HHLT-gal4*. Screen and line identifiers are shown along with the targeted gene for knockdown by RNAi, the associated whole-animal lymph gland phenotype based upon GFP expression, and the Peroxidasin (Pxn) expression phenotype of dissected lymph glands.

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
