## [Decision Letter]

Thank you for sending your work entitled “Maintenance of *Drosophila* blood progenitors by equilibrium signal regulators functioning in differentiating blood cells” for consideration at *eLife*. Your article has been favorably evaluated by K (Vijay) VijayRaghavan (Senior editor) and 3 reviewers, one of whom served as a guest Reviewing editor.

The Reviewing editor and the other reviewers discussed their comments before we reached this decision, and the Reviewing editor has assembled the following comments to help you prepare a revised submission.

This paper from Mondal and colleagues addresses how blood cell progenitors within the Drosophila larval lymph gland are maintained. The work follows on from an earlier Cell paper that showed the existence of an 'equilibrium signal' from differentiating blood cells centered around PVR. In this study the authors have carried out a well-designed screen to identify novel genes required for blood progenitor maintenance and successfully identify 20 novel regulators of this process. The study then focuses on those that regulate the 'equilibrium signal' in the differentiating cells, namely bip1 and Nup98. The upstream regulation of PVR is an understudied and very important question that has wide-reaching implications, which will likely benefit our understanding of the regulation of this receptor family across species.

Technically this paper is excellent, the screen is beautifully designed and the HHLT-gal4 line is likely to be a great resource to the research community. However, there is little follow up work to the screen. Whilst the screen must have been very labour intensive the paper does feel more like a 'technique paper' rather than a ground-breaking research paper. Therefore, the storyline of the manuscript would benefit from re-shaping and focusing, placing less emphasis on the early steps of screening and more on PVR regulation. The authors may even want the title to reflect this. Some experimental clarifications and amendments of the text would be required to make the manuscript suitable for publication.

Substantive concerns:

1) The authors should address whether regulation of PVR by Nup98, bip1 and RpS8 is cell autonomous, or related to non-autonomous events, which could even originate from the population of tissue hemocytes/macrophages (by the authors called 'circulating hemocytes', see comment below), which express the same drivers as differentiating/-ed hemocytes of the LG (ie Hml-GAL4 and Pxn-GAL4). This overlap in expression is in fact a major caveat of most LG papers, an issue that receives increasing attention now that the biology of larval tissue hemocytes, the active hemocyte population of the larva, has been revealed in more detail (see below). The LG field would greatly benefit from an expression system for differentiated LG-hemocytes only (e.g. by inducible elimination of GAL4 in the earlier differentiated population of tissue hemocytes). However, in the absence of such a tool the authors need to retreat to other methods to demonstrate whether or not the presented regulation of PVR is cell autonomous. These methods should comprise:

1a) Use of flipout-GAL4 to generate *in vivo* RNAi clones of the genes in question, assessing PVR expression and the GAL4 expressing clones (e.g. by UAS-GFP) in the LG. Side-by-side comparison of kd and control tissue should allow to evaluate cell autonomous effects.

1b) Use of Hand-GAL4 with UAS-GAL4, or HHLT without Hml-GAL4, as driver to determine the effects of RNAi of the genes in question on LG PVR expression, in the absence of expression in tissue hemocytes.

2) The authors show all genes of the RNAi screen that led to increased Pxn expression. Do all or just some of these kds affect PVR expression? Was this a selection criterion by the authors to go forward with specific genes?

3) What is the overexpression phenotype of Nup98, bip1 and RpS8? From the phenotypes listed in [Supplementary-material SD1-data] it looks as if they may not be fully complementary to the RNAi phenotypes. This is not a big issue as complex regulatory effects are possible, but nevertheless the authors should mention phenotypes and expression levels of PVR.

4) The discussion mainly focuses on specific hypotheses, i.e. a functional transcription complex consisting of bip1 and RpL8, suggested by third party interaction reports, and Nup98 targeting the promoter of PVR and other genes by third party ChIP data. While these are possible mechanisms, the authors should also discuss alternative and more traditional scenarios, such as general roles of nuclear pore components and consequences of their loss-of-function, and a role of ribosomal subunits either as part of the general translation machinery or in the translation regulation of specific transcripts (see e.g. review by Xue and Barna Nat Rev Mol Cell Biol 2012). Do the authors know whether other nup genes or other ribosomal proteins also have an effect on PVR expression in their system? In this context it would be very interesting to find out whether the identified PVR upstream regulators, and maybe related genes of the same functional families, may have been found by other approaches such as genome-wide RNAi screening for PVR modifiers. Any information from complementary systems would greatly synergize with the observations made by the authors.

5) Some phrasing of the manuscript is misleading and should be corrected.

5a) In the introduction, the LG is presented as 'the larval hematopoietic organ'. This may mislead to think that all hemocyte tools have high LG specificity and no crosstalk with another blood cell population may take place. To provide a more balanced view, the population of larval self-renewing tissue hemocytes/macrophages needs to be mentioned, and appropriate literature should be cited (Markus PNAS 2009; Makhijani et al. Development 2011; Makhijani and Brückner Fly 2012; Gold and Brückner Exp Hematology 2014).

5b) Related to this, when using Hml-GAL4 or another driver active in both the LG and tissue hemocyte populations, the authors should stay away from overstating specificity such as “overexpression of bip1 in Nup98 RNAi lymph glands...” (main text discussing Figure 6).

5c) Hml-GAL4 does not only express in 'differentiating' hemocytes, but mainly in fully 'differentiated' hemocytes of both the tissue hemocyte population and differentiated LG hemocytes. The term 'differentiating' should therefore be used with caution.

5d) 'Circulating hemocytes' should be changed to 'tissue hemocytes', which are also known as 'larval hemocytes'. Recent research has shown that this independent population of hemocytes is largely resident in inductive tissue microenvironments (Hematopoietic Pockets), and under unchallenged conditions enters circulation only gradually and rather late in larval life (see references above).

---

## [Author Response]

*Technically this paper is excellent, the screen is beautifully designed and the HHLT-gal4 line is likely to be a great resource to the research community. However, there is little follow up work to the screen. Whilst the screen must have been very labour intensive the paper does feel more like a 'technique paper' rather than a ground-breaking research paper. Therefore, the storyline of the manuscript would benefit from re-shaping and focusing, placing less emphasis on the early steps of screening and more on PVR regulation. The authors may even want the title to reflect this. Some experimental clarifications and amendments of the text would be required to make the manuscript suitable for publication*.

Thank you very much to the reviewers for the positive comments and constructive criticism. As suggested by the reviewers, the text of the manuscript has now been reshaped to focus less on the early screening steps. In response to specific reviewer concerns about the description of the screen (below), we have removed the proof-of-principle details of the screen from Figure 2 (to a child figure and legend for those that would like to see this part of the screen) and have streamlined the text accordingly. Additionally, we have changed the title to reflect the central role of PVR and its regulation.

*Substantive concerns*:

*1) The authors should address whether regulation of PVR by Nup98, bip1 and RpS8 is cell autonomous, or related to non-autonomous events, which could even originate from the population of tissue hemocytes/macrophages (by the authors called 'circulating hemocytes', see comment below), which express the same drivers as differentiating/-ed hemocytes of the LG (ie Hml-GAL4 and Pxn-GAL4). This overlap in expression is in fact a major caveat of most LG papers, an issue that receives increasing attention now that the biology of larval tissue hemocytes, the active hemocyte population of the larva, has been revealed in more detail (see below). The LG field would greatly benefit from an expression system for differentiated LG-hemocytes only (e.g. by inducible elimination of GAL4 in the earlier differentiated population of tissue hemocytes). However, in the absence of such a tool the authors need to retreat to other methods to demonstrate whether or not the presented regulation of PVR is cell autonomous. These methods should comprise*:

*1a) Use of flipout-GAL4 to generate in vivo RNAi clones of the genes in question, assessing PVR expression and the GAL4 expressing clones (e.g. by UAS-GFP) in the LG. Side-by-side comparison of kd and control tissue should allow to evaluate cell autonomous effects*.

*1b) Use of Hand-GAL4 with UAS-GAL4, or HHLT without Hml-GAL4, as driver to determine the effects of RNAi of the genes in question on LG PVR expression, in the absence of expression in tissue hemocytes*.

For 1a and 1b, the major issue is whether the observed PVR phenotype in the lymph gland is due to autonomous RNAi effects or due to non-autonomous, secondary effects of RNAi in circulating/sessile, non-lymph gland hemocytes. We have addressed both 1a and 1b by expressing RNAi using Hand-gal4, which is expressed in the lymph gland but not in circulating or sessile hemocytes. Small flipout-gal4 clones (UAS-FLP Ay-gal4 UAS-GFP) constitutively driving RNAi were made at low temperature (room temp and 25 °C), while larger clones covering the most or all of the lymph gland were generated at 29 °C. In both cases, PVR expression was assessed in RNAi backgrounds. In both cases, the evidence indicates a lymph gland-autonomous role in the regulation of PVR. Clones expressing RNAi to PVR show a robust knockdown of PVR protein expression, whereas mock clones do not. RNAi knockdown of bip1 and Nup98 strongly and moderately reduces PVR protein expression, respectively (Figure 6—figure supplement 2). By contrast, clones expressing RNAi for RpS8 show little if any effect on PVR expression (not shown). Why RpS8 RNAi fails to give a PVR phenotype is unclear, however it is likely that the regulation of PVR by these factors is complex and that the generation of small clones is not equivalent to expressing RNAi throughout a defined cell population (e.g., the Hml-gal4-expressing cells of the CZ). More importantly, to address whether bip1, Nup98, or RpS8 knockdown in the circulating/sessile cell population can affect lymph gland PVR expression, we used srpHemo-gal4 (Bruckner et al., 2004) as an RNAi driver, which expresses in a large fraction of third-instar circulating cells but with essentially no expression in lymph glands. RNAi for bip1, Nup98, and RpS8 were used and for each PVR expression in the lymph gland was unaffected (Figure 6—figure supplement 2).

*2) The authors show all genes of the RNAi screen that led to increased Pxn expression. Do all or just some of these kds affect PVR expression? Was this a selection criterion by the authors to go forward with specific genes*?

PVR expression was not a primary screening criterion, so the RNAi lines were not assessed as a group. However, we do know that not all RNAi knockdowns cause a reduction or loss of PVR expression, because CTPsynthase RNAi exhibits normal PVR levels (see Results and Discussion section). PVR expression was only assessed after three RNAi lines (targeting bip1, Nup98, and CTPsynthase) were identified that cause a loss of progenitors when expressed with Hml-gal4 (Figure 4, similar to RNAi of PVR/loss of equilibrium signaling; Mondal et al., 2011) in differentiating/mature cells, and after the bip1RNAi phenotype could be suppressed by expressing PVR (Figure 6).

*3) What is the overexpression phenotype of Nup98, bip1 and RpS8? From the phenotypes listed in*
[Supplementary-material SD1-data]
*it looks as if they may not be fully complementary to the RNAi phenotypes. This is not a big issue as complex regulatory effects are possible, but nevertheless the authors should mention phenotypes and expression levels of PVR*.

Overexpression of bip1, Nup98, or RpS8 has no significant effect on PVR levels (Figure 6—figure supplement 5).

*4) The discussion mainly focuses on specific hypotheses, i.e. a functional transcription complex consisting of bip1 and RpL8, suggested by third party interaction reports, and Nup98 targeting the promoter of PVR and other genes by third party ChIP data. While these are possible mechanisms, the authors should also discuss alternative and more traditional scenarios, such as general roles of nuclear pore components and consequences of their loss-of-function, and a role of ribosomal subunits either as part of the general translation machinery or in the translation regulation of specific transcripts (see e.g. review by Xue and Barna Nat Rev Mol Cell Biol 2012)*.

We agree with the reviewers’ comments and have now revised and expanded our discussion of this issue in the text and have included the appropriate references.

Do the authors know whether other nup genes or other ribosomal proteins also have an effect on PVR expression in their system? In this context it would be very interesting to find out whether the identified PVR upstream regulators, and maybe related genes of the same functional families, may have been found by other approaches such as genome-wide RNAi screening for PVR modifiers. Any information from complementary systems would greatly synergize with the observations made by the authors.

This is an important question. We found that unlike Nup98, loss of the nucleoporin Sec13 does not cause a loss of Pvr (Figure 6—figure supplement 3). In response to the reviewer’s question, we have now used RNAi to knock down 3 additional nucleoporins (Nup154, 214, and 358; using Hml-gal4) and found that they do not affect PVR levels (Figure 6—figure supplement 5). Thus, the function of Nup98 is specific. We have not analyzed additional ribosomal protein genes since there are a large number of genes that belong to this family.

During our analysis we looked at the literature for possible connections between nucleoporins or ribosomal protein and PVR, but did not find any reports connecting them beyond what we cite in the manuscript already. We are not aware of any previous screen to identify regulators of PVR; however, as mentioned in the manuscript, a genome-wide RNAi screen in S2 cells that identified PVR as a major regulator of proliferation also identified RpS8 (Sims et al., 2009), although no attempt was made by the authors to link it to PVR regulation.

*5) Some phrasing of the manuscript is misleading and should be corrected*.

*5a) In the introduction, the LG is presented as 'the larval hematopoietic organ'. This may mislead to think that all hemocyte tools have high LG specificity and no crosstalk with another blood cell population may take place. To provide a more balanced view, the population of larval self-renewing tissue hemocytes/macrophages needs to be mentioned, and appropriate literature should be cited (Markus PNAS 2009; Makhijani et al. Development 2011; Makhijani and Brückner Fly 2012; Gold and Brückner Exp Hematology 2014)*.

We have added text to state that multiple hematopoietic populations are evident in Drosophila. We have also cited the appropriate literature, however we do want to emphasize that this manuscript deals only with the hematopoietic compartments found within the lymph gland.

*5b) Related to this, when using Hml-GAL4 or another driver active in both the LG and tissue hemocyte populations, the authors should stay away from overstating specificity such as “overexpression of bip1 in Nup98 RNAi lymph glands...” (main text discussing*
Figure 6*)*.

To say that Hml-gal4 is used to overexpress UAS constructs in the lymph gland is not incorrect; however we have clarified that Hml-gal4 expression is not limited to the lymph gland (see 5c below) at its first description in the manuscript. For the sake of simplicity, and, given our experimental results (including those conducted here to demonstrate autonomy), we have kept the original language elsewhere since Hml-gal4-expressing cells in the lymph gland are the relevant cells.

*5c) Hml-GAL4 does not only express in 'differentiating' hemocytes, but mainly in fully 'differentiated' hemocytes of both the tissue hemocyte population and differentiated LG hemocytes. The term 'differentiating' should therefore be used with caution*.

This point is correct. Hml-gal4 activity is a definitive marker for the earliest stages of differentiation in the lymph gland (Figure 6 and Jung et al., 2005) but it is also expressed by more mature cells in the lymph gland (such as those expressing the P1 marker) and those cells in circulation and sessile populations. We have rechecked the manuscript and have clarified this point where appropriate.

*5d) 'Circulating hemocytes' should be changed to 'tissue hemocytes', which are also known as 'larval hemocytes'. Recent research has shown that this independent population of hemocytes is largely resident in inductive tissue microenvironments (Hematopoietic Pockets), and under unchallenged conditions enters circulation only gradually and rather late in larval life (see references above)*.

In consideration of established and newer contributions to the literature (Rizki, 1978; Zetterval et al., 2004; Markus et al., 2009; Honti et al., 2010; Makhijani et al., 2011, among others) we have chosen to use “circulating and sessile cells” in the manuscript to refer to non-lymph gland larval hemocytes. These terms are well established and widely used, including the view that circulating/sessile cells and lymph gland cells represent distinct hemocyte populations arising from their different developmental origins. Perhaps there is need for new nomenclature that combines the sessile and circulating pool. However neither “tissue hemocyte” nor “larval hemocyte” is appropriate for them as the hemocytes within the lymph gland are contained within a larval tissue and released into the hemocoel both upon infection and at the end of the larval period.